# Systematic review and meta-analysis of the effectiveness and perinatal outcomes of COVID-19 vaccination in pregnancy

Smriti Prasad[1,15], Erkan Kalafat[2,3,15], Helena Blakeway [1], Rosemary Townsend[4,5], Pat O'Brien[6,7], Edward Morris[6,8], Tim Draycott[6,9], Shakila Thangaratinam[10], Kirsty Le Doare[11], Shamez Ladhani[12,13], Peter von Dadelszen[14], Laura A. Magee[14], Paul Heath[11] & Asma Khalil [1,5✉]

Safety and effectiveness of COVID-19 vaccines during pregnancy is a particular concern affecting vaccination uptake by this vulnerable group. Here we evaluated evidence from 23 studies including 117,552 COVID-19 vaccinated pregnant people, almost exclusively with mRNA vaccines. We show that the effectiveness of mRNA vaccination against RT-PCR confirmed SARS-CoV-2 infection 7 days after second dose was 89·5% (95% CI 69·0-96·4%, 18,828 vaccinated pregnant people, $I^2 = 73·9\%$). The risk of stillbirth was significantly lower in the vaccinated cohort by 15% (pooled OR 0·85; 95% CI 0·73-0·99, 66,067 vaccinated vs. 424,624 unvaccinated, $I^2 = 93·9\%$). There was no evidence of a higher risk of adverse outcomes including miscarriage, earlier gestation at birth, placental abruption, pulmonary embolism, postpartum haemorrhage, maternal death, intensive care unit admission, lower birthweight Z-score, or neonatal intensive care unit admission ($p > 0.05$ for all). COVID-19 mRNA vaccination in pregnancy appears to be safe and is associated with a reduction in stillbirth.

[1] Fetal Medicine Unit, St George's Hospital, St George's University of London, London, UK. [2] Department of Statistics, Faculty of Arts and Sciences, Middle East Technical University, Ankara, Turkey. [3] Department of Obstetrics and Gynaecology, School of Medicine, Koc University, Istanbul, Turkey. [4] Usher Institute of Population Health Sciences and Informatics, University of Edinburgh, Edinburgh, UK. [5] Vascular Biology Research Centre, Molecular and Clinical Sciences Research Institute, St George's University of London, London, UK. [6] The Royal College of Obstetricians and Gynaecologists, London, UK. [7] University College London Hospitals NHS Foundation Trust, London, UK. [8] Norfolk and Norwich University Hospitals NHS Foundation Trust, Norwich, Norfolk, UK. [9] North Bristol NHS Trust Department of Women's Health, Westbury-on-Trym, Bristol, UK. [10] Institute of Metabolism and Systems Research, WHO Collaborating Centre for Women's Health, University of Birmingham, Birmingham, UK. [11] Paediatric Infectious Diseases Research Group and Vaccine Institute, Institute of Infection and Immunity, St George's University of London, London, UK. [12] Immunisation and Countermeasures Division, Public Health England, England, UK. [13] British Paediatric Surveillance Unit, Royal College of Paediatrics and Child Health, England, UK. [14] Institute of Women and Children's Health, School of Life Course Sciences, King's College London, London, UK. [15] These authors contributed equally: Smriti Prasad, Erkan Kalafat. ✉email: akhalil@sgul.ac.uk

With the first COVID-19 vaccination trials reporting the effectiveness of vaccination against COVID-19 in December 2020[1,2], mass vaccination started immediately in higher income countries and has progressed at unprecedented pace, albeit with disappointing variation in coverage, locally and globally[3]. One under-vaccinated group is pregnant people. Exclusion of pregnant people from initial COVID-19 vaccine trials, lack of experience with mRNA vaccine platforms outside research settings in this group, and resultant variable and ambiguous guidance on vaccination from official and professional bodies, as well as antivaccine disinformation, contributed to vaccine hesitancy among pregnant people[4–8].

SARS-CoV-2 infection in pregnancy can have devastating effects, with evidence showing increased rates of admission to hospital and intensive care units (ICU), maternal death, stillbirth, pre-eclampsia and preterm birth[9,10]. In the UK, the rate of hospital and ICU admission and the associated co-morbidities has increased with each wave[11]. Data from the UK Obstetric Surveillance System (UKOSS) indicate that the overwhelming majority of pregnant people who required hospitalization or ICU care for COVID-19 during the delta wave were unvaccinated[12]. With the emergence of the Omicron variant, the pandemic appears to be far from over and the urgent need for vaccination of pregnant people cannot be overemphasised.

Emerging data from individual observational studies and large case series are consistent with pre-clinical studies that suggested that COVID-19 vaccines have no adverse effects on pregnancy or neonatal outcomes[13–17]. Clinical trials are underway to investigate the outstanding questions about COVID-19 vaccination in pregnancy, including the optimal dosing schedule, and the duration and efficacy of antibodies transferred to the neonate transplacentally and in breastmilk[18]. There is an immediate need for high-quality robust information to support pregnant people considering COVID-19 vaccination, pending additional updates from large national registries and the results of ongoing trials. We conducted a systematic review and meta-analysis of published data on the effects of COVID-19 vaccination in pregnancy, and on vaccine effectiveness in pregnancy.

## Results

Of 578 abstracts screened, 54 were relevant for full-text review. Included were 18 studies[8,13,19–34] and 5 randomised trials[1,2,35–37], reporting on 117,552 people vaccinated during pregnancy (Fig. 1). Studies excluded at full text review and their reasons for exclusion are shown in Supplementary Table 1.

Supplementary Table 2 shows the characteristics of the 23 included studies. Fifteen studies[8,19–24,27–34] compared results between COVID-19 vaccinated and unvaccinated persons and were included in the meta-analysis; three of these studies compared outcomes between vaccinated pregnant persons and non-pregnant women[13,25,26]. Five randomised trials reported on inadvertent exposure to COVID-19 vaccine during pregnancy[1,2,35–37]. Included studies reported data from six countries: Israel, USA, UK, Norway, Qatar and Canada. Some studies included data on both mRNA and viral vector vaccines[8,21,26–30,32,33]. Most did not report outcomes according to vaccine type, trimester at vaccination or number of doses. Also, with one exception[19], studies did not provide data regarding prior infection status. Tables 1 and 2 show the effectiveness of COVID-19 vaccination in pregnancy, and the impact on pregnancy outcomes (vs. no vaccination).

### Effectiveness of vaccination against SARS-CoV-2 infection in pregnant people. Three observational studies[22–24] examined the effectiveness of mRNA vaccination against SARS-CoV-2 infection, comparing 18,828 vaccinated pregnant people and 18,828 unvaccinated pregnancy controls. While other studies reported on SARS-CoV-2 infection[20,30] as an outcome measure, the duration of follow-up and accounting strategy of time-varying confounding were unclear, so they were not included in the quantitative summary.

Two of the three included studies matched for demographic and clinical characteristics[23,24], while the other matched only for maternal age[22]. All three studies adjusted for known confounders and were at moderate risk of bias by ROBINS-I (Supplementary Table 3). Dagan et al.[23] and Goldshtein et al.[24] both used national-level data in Israel; the sampling period coincided with a pandemic peak for Goldshtein et al. (December 2020–February 2021), while Dagan et al. also captured the time period following the end of the same peak (December 2020–June 2021). Butt et al.[22] included a population in Qatar and the sampling period included a peak and a low-case period preceding it (December 2020–May 2021). Both Israel and Qatar recommend a 21-day dosing interval for mRNA vaccines, and follow-up in the included studies started one week after the second dose and ranged from 42 to 50 days.

The effectiveness of mRNA vaccination against RT-PCR confirmed SARS-CoV-2 infection 7 days after the second dose was 89.5% (95% CI: 69.0–96.4%, $I^2 = 73.9\%$) (Table 1, Fig. 2). There was significant between-study heterogeneity in the magnitude of effectiveness, in particular effectiveness shown when the interval after peak infection was included[23].

### Pairwise meta-analysis of maternal and perinatal outcomes. Of 18 observational studies of the impact of COVID-19 vaccination (vs. no vaccination) on pregnancy outcomes, the vast majority were of mRNA vaccines, with very few of viral vector vaccines[8,21,26–30,33]. Twelve studies were observational[8,19,20,22–24,27,29–32,34] and three were governmental reports[21,28,33]. We did not include two studies reporting vaccination side-effects[13,26] and a single study[25] reporting antibody levels, as there was no pregnancy control group. Three studies were at overall serious risk of bias[19,20,32], with the remainder at moderate risk[8,13,21–34] (Supplementary Tables 3 and 4). No study was deemed to be at risk of bias for classification of interventions or deviation from intended interventions domains. Problematic missing outcome data were observed in one study[19].

Table 2 shows the impact of COVID-19 vaccination on pregnancy outcomes. By pair-wise meta-analysis of outcomes among vaccinated (vs. unvaccinated) pregnant people, there was no increase in any adverse outcome examined, for the mother or baby. In fact, there was some evidence of benefit. Neither funnel plot asymmetry testing nor subgroup analysis, by vaccine type or gestational age at vaccination, was possible, due to an inadequate number of relevant studies.

A 15% decrease in the odds of stillbirth was associated with vaccination (vs. no vaccination) in pregnancy (pooled OR 0.85; 95% CI 0.73–0.99, 7 studies, 66,067 vaccinated vs. 424,624 unvaccinated, $P = 0.035$, $I^2 = 93.9\%$, Fig. 3) (Table 2). Substantial statistical heterogeneity was observed attributable to three reports[24,30,31]. The stillbirth rates reported by Morgan et al.[30] and Goldshtein et al.[24] were unusually low (<1:1000) and may have overestimated the protective effect due to zero and one events observed in the vaccinated group, respectively. The study by Rottenstreich et al.[31] was at risk of confounding bias as vaccinated pregnant people were older, had higher rates of previous miscarriage, fertility treatments and diabetes.

Hypoxic brain injury, labelled as 'hypoxic ischaemic encephalopathy' and 'birth asphyxia' (each undefined further), was reduced in odds by 71% in association with vaccination (vs. no vaccination) in pregnancy (pooled OR 0.29; 95% CI 0.09–1.00, 2 studies[31,32] 852 vaccinated vs. 2925 unvaccinated, $P = 0.049$,

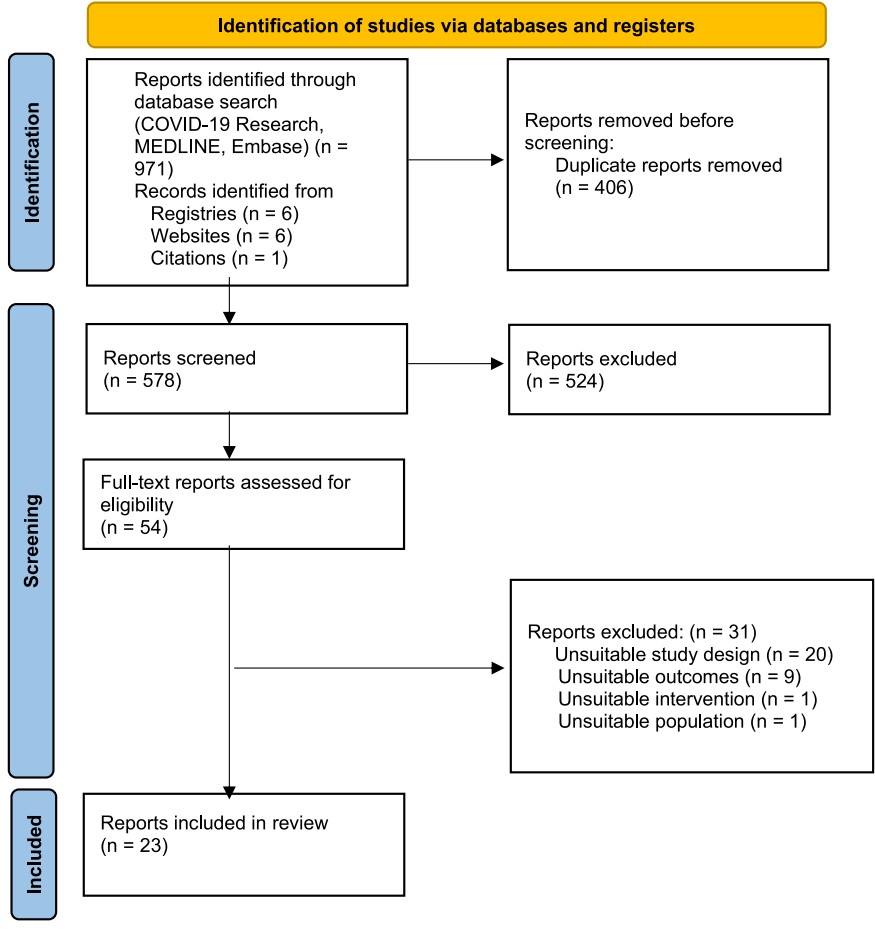

**Fig. 1 PRISMA flow diagram.** Flow diagram summarises the number of studies excluded at each stage.

**Table 1 Meta-analysis of maternal and perinatal outcomes in studies comparing COVID-19 vaccinated and unvaccinated pregnancies.**

| Outcome | Studies (n) | Vaccinated (n) | Unvaccinated (n) | Effect estimate[a] | P | I² (%) |
|---|---|---|---|---|---|---|
| Vaccine effectiveness | 3[22–24] | 18,828 | 18,828 | 89.5% (69.0 – 96.4%) | <0.0001 | 73.9 |
| HDP | 3[31,32,34] | 1765 | 6411 | 1.09 (0.84 – 1.41) | 0.499 | 0.0 |
| Pre-eclampsia | 3[19,24,32] | 7756 | 9454 | 0.94 (0.53 – 1·66) | 0.826 | 0.0 |
| Placental abruption | 3[8,31,34] | 1758 | 4948 | 0.57 (0.32–1.11) | 0.105 | 0.0 |
| Pulmonary embolism | 2[24,32] | 7670 | 9392 | 0.34 (0.0–78.2) | 0.698 | NA |
| Postpartum haemorrhage | 4[8,31,32,34] | 1898 | 6810 | 0.89 (0.5–1.44) | 0.634 | 20.4 |
| ICU admission | 4[8,30–32] | 2317 | 12,084 | 1.45 (0.66–3.16) | 0.356 | 0.0 |
| Maternal death | 6[8,22–24,30,32] | 20,403 | 29,819 | 0.32 (0.0–103.4) | 0.696 | NA |

All P values are two-sided without any adjustments. Effects estimates were pooled using one of the following estimators (maximum-likelihood, restricted maximum-likelihood, inverse variance).
HDP hypertensive disorders of pregnancy, ICU intensive care unit admission, n number.
[a]Effect estimates are reported as odds ratio (binary outcomes) and vaccine effectiveness (1-hazard ratio) and 95% confidence intervals.

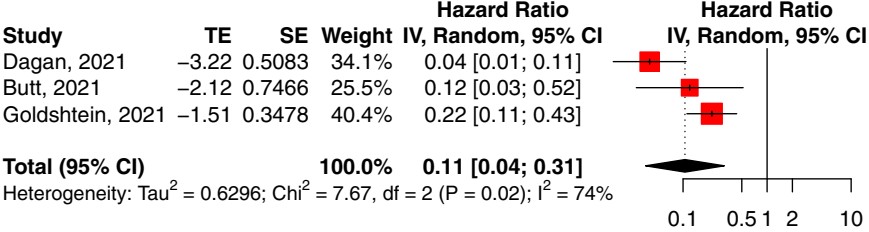

**Fig. 2 Forest plot of studies reporting vaccine effectiveness.** Vertical ticks within the red boxes and horizontal lines show the mean effect and 95% confidences interval for each study. Black diamond at the bottom shows the cumulative effect with 95% confidence intervals.

**Table 2 Meta-analysis of maternal and perinatal outcomes in studies comparing COVID-19 vaccinated and unvaccinated pregnancies.**

| Outcome | Studies (n) | Vaccinated (n) | Unvaccinated (n) | Effect estimate* | P | $I^2$ (%) |
|---|---|---|---|---|---|---|
| Miscarriage | 2[27,29] | 15,684 | 108,249 | 1.00 (0.92– 1.09) | 0.988 | 19.8 |
| Fetal anomalies | 2[8,20] | 335 | 523 | 0.91 (0.40– 2.06) | 0.820 | 0.0 |
| Stillbirth | 7[8,21,24,30-33] | 66,067 | 425,624 | 0.85 (0.73–0.99) | 0.035 | 93.9 |
| GA at birth | 4[8,24,31,34] | 3145 | 6375 | −0.04 (−0.30 to 0.29) | 0.966 | 85.3 |
| PTB < 37 weeks' gestation | 6[8,24,28,31-33] | 37,195 | 369,924 | 0.89 (0.73– 1.09) | 0.269 | 96.8 |
| PTB < 34 weeks' gestation | 2[8,31] | 845 | 1462 | 0.88 (0.53– 1.48) | 0.634 | 0.0 |
| PTB < 32 weeks' gestation | 3[8,32,33] | 25,032 | 331,419 | 0.81 (0.34–1.92) | 0.627 | 56.6 |
| PTB < 37 weeks' gestation adjusted for time-varying confounding | 2[8,28] | 10,197 | 36,414 | 0.90 (CI 0.81–1.00) | 0.051 | 0.0 |
| Low BW (<2500 g) | 2[32,33] | 24,899 | 331,020 | 0.99 (0.93–1.04) | 0.621 | 0.0 |
| SGA at birth | 4[8,28,31,34] | 10,686 | 36,647 | 1·00 (0·93 – 1·08) | 0.918 | 16.9 |
| BW, Z-score | 2[8,19] | 219 | 461 | 0.0 (−0.17 to 0.18) | 0.956 | 0.0 |
| BW, grams | 3[24,31,34] | 3012 | 5976 | −5.88 (−28.8 to 16.0) | 0.598 | 0.0 |
| Asphyxia | 2[31,32] | 852 | 2925 | 0.29 (0.09–1.00) | 0.049 | 0.0 |
| 5-min Apgar score < 7 | 3[31,32,34] | 1765 | 6411 | 0.78 (0.37– 1.61) | 0.499 | 46.7 |
| NICU admission | 3[8,31,32] | 985 | 3324 | 0.94 (0.63– 1.40) | 0.764 | 0.0 |

All P values are two-sided without any adjustments. Effects estimates were pooled using one of the following estimators (maximum-likelihood, restricted maximum-likelihood, inverse variance)
BW birthweight, GA gestational age, n number, NICU neonatal intensive care unit, PTB preterm birth, SGA small for gestational age
aEffect estimates are reported as odds ratio (binary outcomes), mean difference (continuous outcomes), hazard ratio (preterm birth adjusted for time varying confounding and 95% confidence intervals.

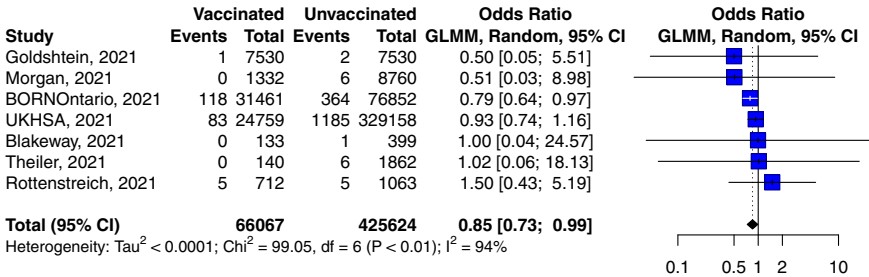

**Fig. 3 Forest plot of studies reporting stillbirth rate.** Vertical ticks within the blue boxes and horizontal lines show the mean effect and 95% confidences interval for each study. Black diamond at the bottom shows the cumulative effect with 95% confidence intervals.

**Fig. 4 Forest plot of studies reporting hypoxic brain injury.** Vertical ticks within the blue boxes and horizontal lines show the mean effect and 95% confidences interval for each study. Black diamond at the bottom shows the cumulative effect with 95% confidence intervals.

$I^2 = 0.0\%$, Fig. 4). However, one included study was at serious risk of bias[32] and estimates were not stable under different meta-analytical methods (Mantel–Haenszel, $P = 0.807$; maximum-likelihood, $P = 0.085$).

There was no significant impact of vaccination (vs. no vaccination) on the odds of preterm birth as reported, before 37 weeks' gestation (37,195 vaccinated vs. 369,924 unvaccinated, $P = 0.269$, $I^2 = 96.8\%$, Fig. 5a), before 34 weeks' (Fig. 5b), or before 32 weeks' gestation (Fig. 5c). Significant between-study heterogeneity in preterm birth before 37 weeks' gestation was attributed primarily to inconsistency in the estimates between the CDC and the UKHSA reports[28,33]. The crude estimates from these reports did not take into consideration time-varying confounding or the baseline risk of people who opted to be vaccinated during pregnancy. Lipkind et al. (CDC report)[28] and Blakeway et al.[8] reported hazard ratio estimates after adjusting for confounding effects. Blakeway et al.[8] originally reported a hazard ratio censored

after 39 weeks' gestation and we re-estimated the hazard ratio for 37 weeks using the original raw data, so it can be comparable to Lipkind et al.[28] The cumulative effect shows a non-statistically significant 10% reduction in preterm birth before 37 weeks' gestation following COVID-19 vaccination (pooled HR: 0.90, 95% CI: 0.81–1.00, $I^2 = 0.0\%$, $P = 0.051$). No study distinguished spontaneous from iatrogenic preterm birth.

Although miscarriage rate was reported in multiple studies[20,24,27,29], only two compared vaccinated and unvaccinated populations and accounted for time-varying confounding[27,29]. The pooled odds ratio meta-analysis showed no significant effect of vaccination on miscarriage (pooled OR 1.00; 95% CI 0.92–1.09, 15,684 vaccinated vs. 108,249 unvaccinated population, $P = 0.988$, $I^2 = 19.8\%$, Fig. 6). These findings were consistent with data from five randomised trials[1,2,35–37] that reported miscarriage rates after inadvertent exposure to COVID-19 vaccination in early pregnancy. As the number of reported exposures was small ($N = 4–43$) and the

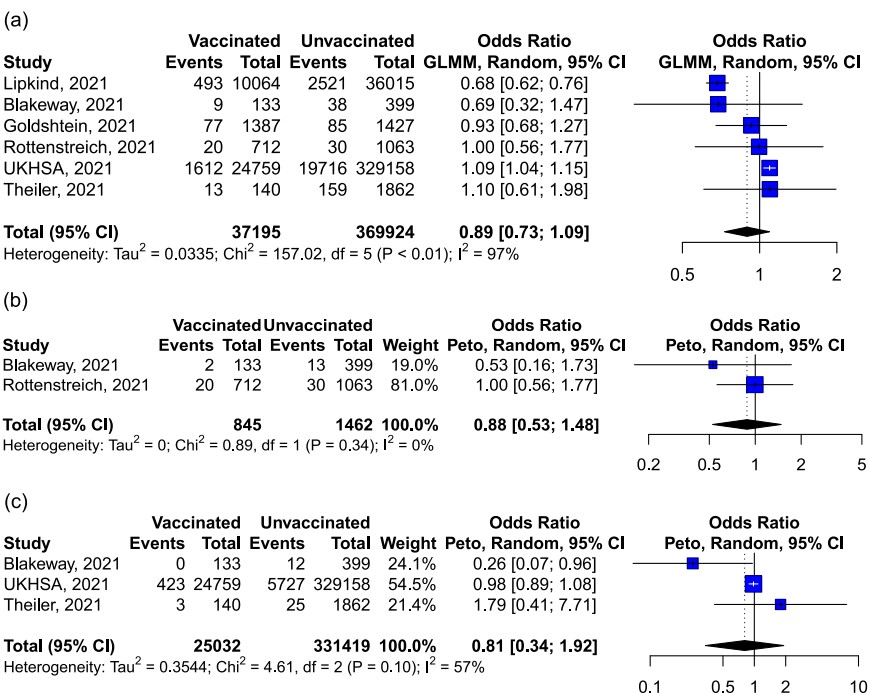

**Fig. 5 Forest plot of studies reporting on preterm birth rate prior to 37 weeks' (a), 34 weeks' (b) and 32 weeks' (c) gestation.** Vertical ticks within the blue boxes and horizontal lines show the mean effect and 95% confidences interval for each study. Black diamond at the bottom shows the cumulative effect with 95% confidence intervals.

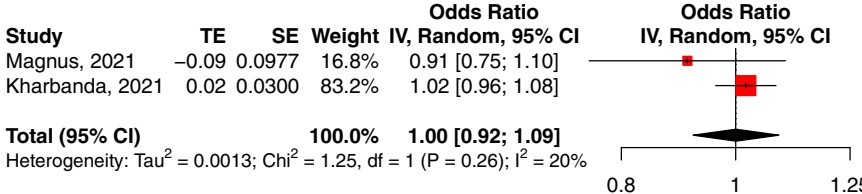

**Fig. 6 Forest plot of studies reporting miscarriage rate and accounting for time-varying confounding.** Vertical ticks within the red boxes and horizontal lines show the mean effect and 95% confidences interval for each study. Black diamond at the bottom shows the cumulative effect with 95% confidence intervals.

vaccine types varied (i.e., two mRNA and three viral vector), quantitative synthesis was not undertaken.

## Discussion

**Summary of key findings.** This systematic review and meta-analysis summarise current data on COVID-19 vaccine effectiveness and safety during pregnancy. The mRNA vaccines appear highly effective against SARS-CoV-2 in pregnancy. The incidence of adverse pregnancy outcomes was not increased among vaccinated compared with unvaccinated pregnancies. In fact, the incidence of stillbirths and possibly, preterm births, was lower among the vaccinated pregnant population. Importantly, there was no increased risk of miscarriage following COVID-19 vaccination, based on data from large national registries and reports of inadvertent exposures in early pregnancy during RCTs.

**Strengths and limitations.** The strengths of this study include its robust methodology and comprehensive literature search, including incorporation of data from the 'grey' literature, such as governmental reports, and data from trials in which pregnant people were inadvertently exposed to the vaccine in early pregnancy. Also, we have included the broadest possible range of reported outcomes, to provide a comprehensive summary of

relevant information to support informed and shared decision-making by pregnant people and maternity care providers.

This study, like most systematic reviews, was limited by the literature on which it was based. Very few studies reported maternal and neonatal outcomes after each dose of the vaccine and according to trimester at vaccination, and there was variation in the outcomes reported, so planned sensitivity analyses could not be performed[38]. Where aggregate meta-analysis was feasible, the conclusions were further challenged by low numbers in the exposed (vaccinated) groups and by the low rates of adverse obstetric outcomes in the countries where published data have emerged. Moreover, even when included studies collected data from pregnant people who received viral vector vaccines, outcome data were not reported according to vaccine type, preventing relevant subgroup analyses. Booster shots gained prominence, especially with the advent of Omicron variant, and most countries allow a mix-and-match approach for booster doses. This increases the importance of safety data on other vaccine types and as well as the effectiveness of booster shots in pregnancy—both significantly lacking in the literature at the time of this review[39]. Finally, we were unable to study the impact of vaccination on spontaneous preterm birth, or vaccine effectiveness in infection-naïve and previously infected subgroups of pregnant people, or according to the COVID-19 variants of concern.

All studies included data from high-income countries, but vaccines are now readily available to pregnant people in many

low-and-middle-income countries. An increasing number of inactivated and viral vector vaccines (Sinovac, Sputnik, Sinopharm, Covaxin, etc.) have been endorsed by the World Health Organization[40] and are increasingly available to pregnant people, particularly in low-and-middle-income countries. Lack of reports from these settings and for these vaccine types represent an important knowledge gap about the safety and efficacy of COVID-19 vaccines in pregnancy.

**Interpretation of the findings**. One question of particular interest was the risk of miscarriage following COVID-19 vaccination in early pregnancy. This is particularly important as up to 40% of pregnancies are unintended and may go unrecognised until 4–8 weeks' gestation or beyond, so that inadvertent vaccination in early pregnancy is likely to be common[41]. The mRNA vaccine causes both antibody and cellular immune responses; given the importance of T-cell suppression in placental development and fetal wellbeing[42], concern has been expressed that the vaccine may increase miscarriage risk. Social media has been full of reports that have fuelled this concern, and many pregnant people have cited this fear as their primary reason for vaccine hesitancy. Our data do not support such concerns, based on both observational data[13,19,24,26,27,29] (even when accounting for potential time-varying confounding[27,29]) and inadvertent exposure in early pregnancy in vaccine trial participants[1,2,35–37].

Our finding of a lower incidence of stillbirths in the vaccinated cohort is important, given that COVID-19 in pregnancy is associated with an increased risk of stillbirth, particularly in the period of Delta dominance[43]. A recent report of population-level data from Scotland found that following SARS-CoV-2 infection in unvaccinated pregnant people, the perinatal death rate was 22.6 per 1000 births, while in contrast, no vaccinated pregnant people with breakthrough infection suffered a perinatal death[44]. Taken together with our findings, these data point to the severity of COVID-19 as the mediator of the COVID-19 and stillbirth relationship. However, the observational nature of the original studies, significant statistical heterogeneity observed in the results and other probable confounders should caution not interpret these results as causal.

Although we noted no significant difference in the crude preterm birth rates, the data do not account for time-varying confounding. For instance, the crude rates from the recent CDC data and UKHSA data are conflicting to the extent that the former suggests a 32% reduction in preterm birth, while the latter suggests a 9% increase. However, the reduced rate in the CDC study is likely explained by the fact that most (~60%) vaccinated people were in their third trimester at the time of vaccination, so many were too late to have a preterm birth. On the other hand, the UK data did not account for the probable higher baseline preterm birth risk of vaccinated people and may have overestimated the effect. When two estimates of hazard ratio (Lipkind et al. [28] and Blakeway et al. [8]) which adjust for time-varying and other possible confounders were combined, we found a non-statistically significant reduction in preterm birth by 10%. This finding is consistent with multiple studies linking COVID-19 infection to preterm birth.

Much of the literature reviewed (other than vaccine trials) was assessed as being at some risk of bias and with substantial between-study differences in some outcomes. Nevertheless, the data reviewed should prove very reassuring to pregnant people and healthcare providers who must make decisions about COVID-19 vaccination in pregnancy now. Further clinical trials will report in 2022. In the interim, prospective registries and active post-marketing surveillance should continue, to build more data about the effectiveness and safety of COVID-19 vaccination in pregnancy, including any rare side-effects that would not be expected to be seen in vaccine trials. In Brazil earlier this year,

there was a report of a maternal death after Astra Zeneca vaccination, but we are unaware of other cases in the published literature[45]. With the viral vector Oxford/Astra Zeneca vaccine, there is a rare risk of vaccine-induced immune thrombotic thrombocytopenia[46,47]; this has prompted some (e.g., the UK, Canada, the USA) but not all countries to withhold this vaccine from people under 40 years of age. These anecdotal reports of very rare complications should not deter the scientific community, healthcare workers and policy makers from disseminating information about the clear benefits of COVID-19 vaccination in pregnancy to both mother and baby.

More recently, reports have emerged of rare post-mRNA vaccination myocarditis, estimated to occur in 2 per million females and 10 per million males aged 18–40 years. Such reactions are typically mild and rapidly self-limiting[48], and occur more commonly in association with COVID-19 infection[49]. We are unaware of such cases in pregnancy.

**Implications for clinical practice**. Worldwide, there has been significant hesitancy among pregnant people to accept COVID-19 vaccination, with one study in the UK finding uptake of only 28.5%—significantly lower than in their non-pregnant peers[8,50–52]. This is likely due to limited evidence on vaccine safety in pregnancy early in the pandemic, and conflicting and changing advice given to pregnant people as the pandemic has evolved[5,53,54]. Our review corroborates the CDC V-Safe registry and MHRA reports, and further emphasizes that COVID-19 vaccination is highly effective in pregnancy to prevent COVID-19, without increasing the risk of adverse pregnancy outcomes. This provides further evidence that the risks of COVID-19 outweigh the rare risks of vaccination in pregnancy, and pregnant people should be encouraged to pursue vaccination, even in the first trimester. However, the lack of high-quality (i.e., low-risk of bias) studies with uniform reporting of clinically important outcomes, as well as under reporting of other vaccines types used in low-to-middle-income countries, are problematic. Studies of other vaccine types are important for improving vaccine coverage in underserved areas. Moreover, a core outcome set for reporting newly developed therapeutics would facilitate uniform high-quality evidence accumulation.

The evidence supports recommendations[55,56] advising COVID-19 vaccination in pregnancy, as it provides maternal and fetal benefit, without increasing perinatal risk. Ongoing post-marketing surveillance, ideally prospective in nature and controlling for bias, is needed to grow the evidence base related to vaccine type, timing in pregnancy, and vaccine effectiveness against newer evolving variants, like Omicron.

## Methods

This systematic review and meta-analysis on the effectiveness and safety of COVID-19 in pregnancy was registered with PROSPERO 2021 (CRD42021274016) and reported according to PRISMA guidelines.

**Study selection and data extraction**. We electronically searched the COVID-19 Research, MEDLINE and Embase databases from 1 December 2020 to 8 September 2021. We updated our search on 13 November 2021 and again on 9 January 2022. For safety reports and briefings, we searched preprint servers including medRxiv; bioRxiv; and national regulatory and advisory body websites, including the Centers for Disease Prevention and Control (CDC), Medicines and Healthcare products Regulatory Agency (MHRA), United Kingdom Health Security Agency (UKHSA, previously known as Public Health England), Federal Drug Administration (FDA), European Medicines Agency (EMA), and Health Canada. The search included relevant medical subject heading terms, keywords, and word variants for pregnancy outcomes, perinatal outcomes, neonatal outcomes, stillbirth, preterm birth, obstetric complications, COVID-19 vaccination and immunization. No language restrictions were applied, and only human studies were included. The search strategy can be found in the supplementary material (Supplementary Table 5). Abstracts and potentially relevant full texts were reviewed independently by two authors (H.B. and S.P.) with any conflicts resolved by consulting a third senior author (A.K.).

We included randomised trials, case-control studies, cohort studies, and data from national registries comparing outcomes between vaccinated (during pregnancy) and unvaccinated pregnancies. Additionally, we retrieved from trial safety reports and briefings, data on outcomes of inadvertent pregnancies occurring during clinical trials. We excluded narrative literature reviews (but explored their reference lists for potentially eligible studies) and reports without contemporary control groups of unvaccinated pregnant people.

The intervention of interest was COVID-19 vaccination in pregnancy, inclusive of all vaccine platforms (mRNA, viral vector, inactivated, etc.). Data were extracted with use of Covidence systematic review software (version 2, Veritas Health Innovation, Melbourne, VIC, Australia) on a pre-designed spreadsheet. The following data were extracted where available: author's name, publication date, study design, country, reported outcome categories, sample size of exposed (vaccinated) and non-exposed (unvaccinated) control population, data collection period, type of vaccine, number of doses received, and timing of administration (trimester of pregnancy). The total number of vaccinated pregnant people and the sum of adverse events in each group were extracted for categorical outcomes. Mean, standard deviation, and the total number of vaccinated pregnant people in each outcome group were extracted for outcomes reported on a continuous scale. We sought to study the following outcomes: (i) confirmed SARS-CoV-2 infection; (ii) maternal outcomes of hypertensive disorders of pregnancy (HDP), pre-eclampsia, placental abruption, pulmonary embolism, postpartum haemorrhage, intensive care unit admission, and maternal death; and (iii) fetal/newborn outcomes of miscarriage, fetal anomalies, stillbirth, gestational age at birth, preterm birth (at <37, <34, and <32 weeks' gestation), birthweight, status at birth, and neonatal intensive care unit admission. For this review, we did not study non-pregnancy-related safety outcomes, reactogenicity or immunogenicity of the candidate COVID-19 vaccines.

**Quality assessment**. Each study was scored according to Cochrane Risk of Bias Tool 2 (randomised) and ROBINS-I (observational), independently by two assessors (E.K., S.P.), and disagreements were resolved by referral to a third reviewer (A.K.).

**Statistical analysis**. Results were reported as summary odds ratios, hazard ratios or mean differences with 95% confidence intervals (CI). Dichotomous outcomes were summarised with a hypergeometric-normal model based on maximum-likelihood estimators[57]. When the event rate prohibited model convergence, approximate likelihood was used to facilitate model convergence. When the number of studies for an outcome was less than three or likelihood-based approaches did not converge, random-effects Mantel–Haenszel or Peto method were preferred depending on event rate. When data extraction for re-estimation of the effect was not possible, study reported estimates (log-odds, log-hazard) and variances were combined directly using generic inverse variance meta-analysis with a restricted maximum-likelihood estimator for between-study variance. Continuous outcome measures were summarised using a random-effects model with restricted maximum likelihood estimation of between study variance[58]. The degree of between study heterogeneity that could not be ascribed to sampling error was explored using the $I^2$ statistic[59], and was interpreted as low ($I^2$: <25%), low to moderate ($I^2$: 25–50%), moderate to substantial ($I^2$: 50–75%), or substantial ($I^2$: >75%). Potential publication bias was assessed using Egger's test and the creation of funnel plots for visual inspection when a sufficient number of studies ($N > 10$) was available. Sensitivity analyses were planned for studies reporting outcomes for single or double dose of vaccination, type of vaccine and trimester at first vaccination. Analyses were conducted using R for statistical computing software (v.4.0.3) and meta package (R Foundation for Statistical Computing, Vienna, Austria).

**Reporting summary**. Further information on research design is available in the Nature Research Reporting Summary linked to this article.

## Data availability
The data used in this study have been deposited in the figshare database [https://figshare.com/articles/dataset/Updated_data_extraction_sheet_COVID_vaccine_18_12_2021_xlsx/19375493].

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

## Author contributions

S.P. and H.B. selected the studies. S.P., H.B. and E.K. extracted the data. R.T. performed the literature search and helped with study selection process. A.K. designed the study. E.K. performed the statistical analyses. S.P., H.B., E.K. and A.K. wrote the initial draft. S.P., H.B., E.K., R.T., P.O.B., E.M., T.D., S.T., K.L.D., S.L., P.V.D., L.A.M., P.H. and A.K. interpreted data and revised the manuscript for important intellectual content. S.P., H.B., E.K., R.T., P.O.B., E.M., T.D., S.T., K.L.D., S.L., P.V.D., L.A.M., P.H. and A.K. gave final approval of the version to be published.

## Competing interests

P.O.B. is co-chair of the Royal College of Obstetricians and Gynaecologists (RCOG) Vaccine Committee. P.O.B. and T.D. are RCOG Vice-Presidents. E.M. is RCOG President. P.H. is a member of the RCOG Vaccine Committee. P.H. is CI of the Preg-Cov trial (UK multicentre COVID vaccination in pregnancy). A.K. is obstetric PI of the Preg-Cov trial (UK multicentre COVID vaccination in pregnancy). A.K. is a member of the COVAX Working Group on COVID vaccination in pregnancy. A.K. is PI of the Pfizer COVID vaccination in pregnancy trial. The remaining authors declare no competing interests.
