## [Peer Review File · Nature Communications]

Systematic review and meta-analysis of the effectiveness and perinatal outcomes of COVID-19 vaccination in pregnancyREVIEWERS' COMMENTS

Reviewer #1 (Remarks to the Author):

This systematic review and meta-analysis provides a summary and overview of the existing literature on COVID-19 vaccination in pregnancy, which looks to be of variable quality with a reasonable amount medium to high risk of bias. This alone is useful information as it highlights the need for more robust original research on this important topic. The lack of evidence regarding non-mRNA vaccines and from low- or middle-income countries is also concerning.

The study has been conducted in a rigorous manner and the authors have followed the relevant guidelines for systematic reviews and meta-analyses. The outcomes reported are highly clinically significant.

The topic is of great international concern, as it is now recognised that pregnant women are higher risk of adverse outcomes from COVID-19, and there has been increased hesitancy among this population due to a combination of antivaccine disinformation and initial mixed messaging from health authorities.

Despite the risks of bias in the original research, the lack of any negative safety signals and hints towards positive pregnancy outcomes (although still at risk of bias) is reassuring and an important message for this population.

The data on effectiveness is likely to be at higher risk of bias due to the non-randomised nature of many of these studies and known differences in behaviours of populations who chose to be vaccinated vs those who do not. However, the results indicating the vaccines to be highly effective should not be a surprise as there is no bioplausible reason they would be substantially less effective in pregnant women, and the evidence for vaccine efficacy outside this population is overwhelming.

I have some comments which I hope might help add to the manuscript in its current format.

Abstract

This is well presented. The findings might benefit from mentioning the risk of bias in the literature and lack of randomised trials. The interpretation section may also benefit from highlighting the need for high quality research in this area, including in low- and middle-income countries and with non-mRNA vaccines.

The above points would also be useful to include in the added value section of the research in context

Introduction

Another contributor to poor vaccine uptake may have been antivaccine disinformation specifically targeted at pregnant women.

Methods

These are well described

Results

These are detailed and comprehensive. It is mentioned that RCT's were included for incidental pregnancies but no report is made of these results (although they are referred to in the discussion).

Discussion

The manuscript correctly highlights the difficulties due to heterogeneity in outcomes and analysis of the observational data here. An implication of this is the importance of standardising post marketing studies within specific populations, such as pregnant women. This ensures higher quality data capture at the individual study level, as well as ensuring meta-analysis can be performed robustly.

There is a reference to CVST because of the AstraZeneca vaccine, however more broadly the concern is thrombocytopaenic thrombosis (which may manifest as CVST or other serious thrombotic events).

In addition, in reference to the myocarditis risk post mRNA vaccine it might be worth highlighting this is a risk which is significantly higher in males, therefore is unlikely to be of major concern to pregnant women.

Implications

The reference to reduction of still birth uses causal language, which I would be cautious of given the biases in these studies. It would be more accurate to state that there was an association with lower rates of stillbirth.

This may also be a useful place to highlight the need for higher quality, standardised post

marketing observational trials, earlier inclusion of pregnant women in clinical trials and studies from LMICs.

Tables and figures

These are presented appropriately and described well in the body of the manuscript.

Reviewer #2 (Remarks to the Author):

This is an important topic that needs to be constantly reviewed as more evidence is published internationally in order to inform vaccine policy.

It remains of very active interest for public health policy and clinician advice to have the maximal amount of high-quality data on the impact of COVID-19 vaccines in pregnancy, particularly with high rates of vaccine hesitancy and misinformation circulating on this topic.

This is a meta-analysis so no original data is produced. There have been other systematic reviews done on this topic to date but given that this is a rapidly moving field through the pandemic, it is reasonable to keep updating or redoing quality data reviews on this sort of topic.

The general conclusions of vaccine general safety and effectiveness are solid and well supported.

The conclusions around miscarriage are reasonable but always challenging due to underreporting of early pregnancy in the observational vaccine studies which were the majority of the studies in this analysis.

The reporting of hypoxic brain injury is interesting and has not been generally seen that there is a potentially a protective effect of the vaccine. I am concerned that the definition of this outcome is highly challenging and was not standardized across studies so I think this conclusion is a stretch and should be softened.

They appropriately acknowledge that they were unable to draw significant conclusions regarding reduction in preterm birth in vaccinated individuals.

The reduced incidence of stillbirth is promising but difficulty in accounting for confounding factors makes this a finding that must be taken with caution, however, the reverse that vaccination was not associated with an increase in stillbirth is very important and perhaps a better way to frame this finding.

Overall, given the heterogeneity of studies on this subject, this remains a valuable contribution to the understanding of the impact of COVID-19 vaccines in pregnancy.

Reviewer #1 Comments	Author response
1. This systematic review and meta-analysis provides a summary and overview of the existing literature on COVID-19 vaccination in pregnancy, which looks to be of variable quality with a reasonable amount medium to high risk of bias. This alone is useful information as it highlights the need for more robust original research on this important topic. The lack of evidence regarding non-mRNA vaccines and from low- or middle-income countries is also concerning. The study has been conducted in a rigorous manner and the authors have followed the relevant guidelines for systematic reviews and meta-analyses. The outcomes reported are highly clinically significant. The topic is of great international concern, as it is now recognised that pregnant women are higher risk of adverse outcomes from COVID-19, and there has been increased hesitancy among this population due to a combination of antivaccine disinformation and initial mixed messaging from health authorities. Despite the risks of bias in the original research, the lack of any negative safety signals and hints towards positive pregnancy outcomes (although still at risk of bias) is reassuring and an important message for this population. The data on effectiveness is likely to be at higher risk of bias due to the non-randomised nature of many of these studies and known differences in behaviours of populations who chose to be vaccinated vs those who do not. However, the results indicating the vaccines to be highly effective should not be a surprise as there is no bioplausible reason they would be substantially less effective in pregnant women, and the evidence for vaccine efficacy outside this population is overwhelming. I have some comments which I hope might help add to the manuscript in its current format.	We thank the reviewer for the positive comments.
2. This is well presented. The findings might benefit from mentioning the risk of bias in the literature and lack of randomised trials. The interpretation section may also benefit from highlighting the need for high quality research in this area, including in low- and middle-income countries and with non-mRNA vaccines. The above points would also be useful to include in the added value section of the research in context	We have highlighted these limitations and the need for high quality research studies as suggested by the Reviewer.

3. Another contributor to poor vaccine uptake may have been antivaccine disinformation specifically targeted at pregnant women.	We have added this information to the Introduction section of the revised manuscript.
4. Methods These are well described	We thank the reviewer for the positive comment.
5. Results These are detailed and comprehensive. It is mentioned that RCT's were included for incidental pregnancies but no report is made of these results (although they are referred to in the discussion).	We mentioned the RCTs at the end of results section "These findings were consistent with data from five randomized trials that reported miscarriage rates after inadvertent exposure to COVID-19 vaccination in early pregnancy. As the number of reported exposures was small (N= 4 to 43) and the vaccine types varied (i.e., two mRNA and three viral vector), quantitative synthesis was not undertaken."
6. Discussion The manuscript correctly highlights the difficulties due to heterogeneity in outcomes and analysis of the observational data here. An implication of this is the importance of standardising post marketing studies within specific populations, such as pregnant women. This ensures higher quality data capture at the individual study level, as well as ensuring meta-analysis can be performed robustly.	We have highlighted this point in the discussion section in the revised version of the manuscript. "However, the lack of high-quality (i.e., low-risk of bias) studies with uniform reporting of clinically important outcomes, as well as under reporting of other vaccines types used in low to middle income countries, are problematic. Studies of other vaccine types are important for improving vaccine coverage in underserved areas. Moreover, a core outcome set for reporting newly developed therapeutics would facilitate uniform high-quality evidence accumulation."
7. There is a reference to CVST because of the AstraZeneca vaccine, however more broadly the concern is thrombocytopaenic thrombosis (which may manifest as CVST or other serious thrombotic events).	We changed the term to "there is a rare risk of vaccine-induced immune thrombotic thrombocytopenia".
8. In addition, in reference to the myocarditis risk post mRNA vaccine it might be worth highlighting this is a risk which is significantly higher in males, therefore is unlikely to be of major concern to pregnant women.	The point has been added to the discussion "More recently, reports have emerged of rare post-mRNA vaccination myocarditis,

	estimated to occur in 2 per million females and 10 per million males aged 18-40 years.”
9. Implications The reference to reduction of still birth uses causal language, which I would be cautious of given the biases in these studies. It would be more accurate to state that there was an association with lower rates of stillbirth.	The reviewer is correct and we cannot infer causality from this data. We have removed the sentence in implications section. “and in fact, reducing the risk of stillbirth” We also highlighted this point in the Discussion section ‘However, the observational nature of the original studies, significant statistical heterogeneity observed in the results and other probable confounders should caution not interpret these results as causal.’
10. This may also be a useful place to highlight the need for higher quality, standardised post marketing observational trials, earlier inclusion of pregnant women in clinical trials and studies from LMICs.	We will emphasize these points in the discussion “However, the lack of high-quality (i.e., low-risk of bias) studies with uniform reporting of clinically important outcomes, as well as under reporting of other vaccines types used in low to middle income countries, are problematic. Studies of other vaccine types are important for improving vaccine coverage in underserved areas. Moreover, a core outcome set for reporting newly developed therapeutics would facilitate uniform high-quality evidence accumulation.”
11. Tables and figures These are presented appropriately and described well in the body of the manuscript.	We needed to move Table 1 to supplementary section and split the table 2 into two sections in accordance with journal guidelines

Reviewer #2 Comments	Author response
1. This is an important topic that needs to be constantly reviewed as more evidence is published internationally in order to inform vaccine policy.	We thank the reviewer for the positive comment.

2. It remains of very active interest for public health policy and clinician advice to have the maximal amount of high-quality data on the impact of COVID-19 vaccines in pregnancy, particularly with high rates of vaccine hesitancy and misinformation circulating on this topic.	We thank the reviewer.
3. This is a meta-analysis so no original data is produced. There have been other systematic reviews done on this topic to date but given that this is a rapidly moving field through the pandemic, it is reasonable to keep updating or redoing quality data reviews on this sort of topic.	We agree with the reviewer.
4. The general conclusions of vaccine general safety and effectiveness are solid and well supported.	We thank the reviewer.
5. The conclusions around miscarriage are reasonable but always challenging due to underreporting of early pregnancy in the observational vaccine studies which were the majority of the studies in this analysis.	We agree with the reviewer. For this reason we included the data on miscarriage from the available RCTs.
6. The reporting of hypoxic brain injury is interesting and has not been generally seen that there is a potentially a protective effect of the vaccine. I am concerned that the definition of this outcome is highly challenging and was not standardized across studies so I think this conclusion is a stretch and should be softened.	We agree with the reviewer and have emphasized the short-comings of this findings. “Hypoxic brain injury, labelled as ‘hypoxic ischemic encephalopathy’ and ‘birth asphyxia’ (each undefined further), was reduced in odds by 71% in association with vaccination (vs. no vaccination) in pregnancy (pooled OR 0.29; 95% CI 0.09 – 1.00, 2 studies^{37,38} 852 vaccinated vs 2,925 unvaccinated, $P = 0.049$, $I^2 = 0.0\%$, Figure 4). However, one included study was at serious risk of bias³⁸ and estimates were not stable under different meta-analytical methods (Mantel-Haenszel, $P = 0.807$; maximum-likelihood, $P = 0.085$).

	“
7. They appropriately acknowledge that they were unable to draw significant conclusions regarding reduction in preterm birth in vaccinated individuals.	We thank the reviewer.
8. The reduced incidence of stillbirth is promising but difficulty in accounting for confounding factors makes this a finding that must be taken with caution, however, the reverse that vaccination was not associated with an increase in stillbirth is very important and perhaps a better way to frame this finding.	We agree with the reviewer and further emphasized the limitations of this finding. 'However, the observational nature of the original studies, significant statistical heterogeneity observed in the results and other probable confounders should caution not interpret these results as causal.'
9. Overall, given the heterogeneity of studies on this subject, this remains a valuable contribution to the understanding of the impact of COVID-19 vaccines in pregnancy.	We thank the reviewer.